# Lyme Disease Biosensors: A Potential Solution to a Diagnostic Dilemma

**DOI:** 10.3390/bios10100137

**Published:** 2020-09-28

**Authors:** Connor Flynn, Anna Ignaszak

**Affiliations:** Department of Chemistry, University of New Brunswick, Fredericton, NB E3B 5A3, Canada; anna.ignaszak@unb.ca

**Keywords:** Lyme disease, biosensor, point-of-care, diagnostics, infectious disease

## Abstract

Over the past four decades, Lyme disease has remained a virulent and pervasive illness, persisting throughout North America and many other regions of the world. Recent increases in illness in many countries has sparked a renewed interest in improved Lyme diagnostics. While current standards of diagnosis are acceptable for the late stages of the disease, it remains difficult to accurately diagnose early forms of the illness. In addition, current diagnostic methods tend to be relatively expensive and require a large degree of laboratory-based analysis. Biosensors represent the fusion of biological materials with chemical techniques to provide simple, inexpensive alternatives to traditional diagnostic methods. Lyme disease biosensors have the potential to better diagnose early stages of the illness and provide possible patients with an inexpensive, commercially available test. This review examines the current state of Lyme disease biosensing, with a focus on previous biosensor development and essential future considerations.

## 1. Introduction

Lyme disease, or Lyme borreliosis, is an increasingly prevalent illness caused by several bacteria in the *Borrelia* genus. While several species can cause Lyme disease, *Borrelia burgdorferi* is the predominant pathogenic species in North America; *B. garinii* and *B. afzelii* are responsible for most infections in Europe and Asia. The transmission of *Borrelia* is facilitated by ticks within the *Ixodes* genus, which serve as the primary vector and reservoir host for the bacterium. *Borrelia* is transferred to a mammalian host when an infected tick attaches to and feeds on the blood of the host. During blood-feeding, *Borrelia,* located in the salivary glands of the tick, enter the mammalian host via the tick’s saliva, transmitting the Lyme bacteria and causing the disease. By exploiting the natural feeding relationship of ticks on mammals, *Borrelia* is easily transmitted to deer, livestock, and humans.

Since Burgdorfer’s seminal paper realizing *Borrelia* as the causative agent of Lyme disease [1], the wide range and inconspicuous nature of the disease have become clear. According to the Centers for Disease Control and Prevention (CDC), there are approximately 30,000 reported cases of Lyme disease in the United States each year; however, more recent estimates place this number as high as 300,000, owing to frequent misdiagnosis and under-reporting [2,3,4]. Not only is Lyme disease the most common vector-borne illness in the US, but recent studies suggest that it is on the rise in other locations as well. With climate change leading to the northern expansion of temperate conditions, reservoir and vector hosts of Lyme disease stray further and further from their original habitats. Because of this, *B. burgdorferi* is expected to expand its territory northward by 250–500 km in the next 30 years [5]. Increases in Lyme disease incidence are already occurring in many parts of Canada and are expected to continue into the next decades [6,7]. As well, Lyme disease continues to rise steadily throughout Europe (e.g., Germany, Sweden, Austria) and is beginning to take hold in Asia, particularly in regions of China [8,9,10].

The earliest manifestation of Lyme disease is the appearance of a typically bullseye-shaped rash, known as an erythema migrans, at the site of infection. While this rash is generally sufficient for Lyme diagnosis, it only occurs in 70–80% of cases, making it unreliable as a main indicator [11]. Other than the erythema migrans, the most common symptoms are headaches and arthralgia, but these are far too general to indicate Lyme disease [12]. As the disease progresses, *Borrelia* disseminate from the site of tick attachment and travel throughout the body, causing early disseminated symptoms that can include multiple erythema migrans, carditis, and meningitis [3]. If left untreated for a prolonged period, Lyme disease may progress to a more severe late stage that can include encephalitis and arthritis, among other serious symptoms. While Lyme disease is easily treated with antibiotics if caught early, delayed diagnosis and/or treatment can prove more difficult to treat and lead to more serious health effects [13].

### 1.1. Current Methods of Diagnosis

In cases where an erythema migrans is not present or proves inconclusive, a global consensus of guidelines recommends the use of a standard two-tiered (STT) serology approach for the diagnosis of Lyme disease [14]. This approach involves an initial enzyme immunoassay (EIA) or immunofluorescence assay (IFA) to measure antibody response to *Borrelia* antigens—often in the form of a whole-cell *Borrelia* sonicate [12]. If this EIA/IFA returns a positive or equivocal result, then a follow-up Western blot is used to verify the presence of antibodies for a panel of specific *Borrelia* proteins. A meta-analysis of thirteen two-tiered serology studies across North America estimated the sensitivity of the approach at 46.3% for those with early-stage Lyme disease (symptoms for less than 30 days), 89.7% for those with early disseminated Lyme disease (30+ days), and 99.4% for those with late-stage Lyme disease [15]. Specificity estimates for the two-tiered methodology were approximately 99% for all disease stages.

Recently, a modified two-tiered (MTT) approach, in which the western blot of the STT is replaced with a second immunoassay, was approved for use in the United States [16]. Compared to the STT, the MTT demonstrates increased sensitivity for early Lyme disease, similar or slightly higher sensitivity for later stages of the disease, and similar specificity for all stages [17]. A comparison of the STT and MTT approaches on several collections of sera from patients with erythema migrans and early Lyme disease estimated the sensitivity of the MTT test to be approximately 50%, while the sensitivity of the STT test on the same samples was approximately 40% [18].

While the sensitivity of the current two-tiered serology approaches is very high for disseminated Lyme disease, it remains quite difficult to detect the disease in its early stages. The development of anti-*Borrelia* antibodies can take upwards of three weeks to reach sufficient detection levels in the blood [19]. Since the current standard relies on these antibodies, it is evident that a more direct method of detection is necessary to diagnose Lyme disease prior to its dissemination. The necessity of improved diagnostics is underscored by the recent and expected increases in Lyme disease, the potential for serious health difficulties, and the current diagnosis difficulties (e.g., misdiagnosis, false positives) [3].

### 1.2. Polymerase Chain Reaction (PCR) and Bacterial Cultures

PCR is a biochemical technique used to amplify DNA sequences, providing millions of copies in a short period of time. Through PCR, scientists can amplify specific DNA samples to detectable levels in a manner analogous to sample preconcentration. PCR has been used to detect Lyme disease in clinical samples (e.g., blood, urine, cerebral spinal fluid) [20]. However, PCR-based tests typically have low sensitivity and are thus not typically used in clinical diagnosis [12]. In addition, *Borrelia* is not effectively detected via PCR in humans due to its highly transient bacteremia phase and low concentrations in the body [21,22]. PCR is commonly employed to confirm the presence of *Borrelia* in ticks and other hosts (e.g., deer), where bacterial concentrations are higher [23,24]. It should be mentioned that PCR has seen many improvements since its introduction, and the development of new clinical PCR-based assays for Lyme disease is an area of active research [25].

Cultures grown from samples suspected of containing *B. burgdorferi* (e.g., skin biopsy from erythema migrans, blood) have also been explored as potential diagnostic tools [26]. In addition to low sensitivity, culture-based methods also fall short due to the slow-growing nature of the Lyme bacteria [12]. Still, *B. burgdorferi* culturing remains important in providing lab-based samples to be used in the development of Lyme-related tools and techniques.

## 2. Biosensors to the Rescue

Biosensors comprise a large and rapidly expanding collection of analytical devices that utilize biological components to detect biological and chemical analytes. All biosensors consist of two main components: A biorecognition element and a transducer. Biorecognition elements include antibodies, aptamers, enzymes, cells, and many other biological entities that interact specifically with a biological target. The role of these elements is to react, either chemically or physically, with an analyte of interest in order to indicate its presence in a sample. Since these biological interactions cannot necessarily be directly quantified, a transducer is required to convert these biological events into signals that can be recorded and analyzed in a coherent manner. Transducers may be optical, mechanical, piezoelectric, electrochemical, or any other method that can produce a measurable response.

In comparison to traditional diagnostic methods (e.g., ELISA, PCR), biosensors aim to provide several key advantages that make them ideal for infectious disease diagnosis. The use of highly specific biorecognition elements ensures biosensors have both high selectivity and sensitivity [27]. The minuscule size of these elements allows functionalized surfaces to possess millions of potential interaction sites, leading to large linear response ranges [28]. The use of relatively fast techniques (e.g., impedance) leads to rapid response times, providing results in as little as a few minutes. Biosensors are also typically low-cost, and many have the potential to be miniaturized into portable devices for in-field use [29].

The development of lightweight, portable biosensors for Lyme diagnosis would be invaluable considering that the disease disproportionately affects rural, heavily forested areas more often; these communities, though frequently exposed to the bacteria, often lack the complex equipment required to detect the disease. The current review aims to highlight the recent advances in the development of diagnostic biosensors for Lyme disease, with the goal of providing a clear outline for the future of Lyme diagnosis.

### 2.1. Microfluidics

Microfluidics describes the use and manipulation of fluids in the microliter and sub-microliter range for applications in nanofabrication, electronics, cell-based screening, and diagnostics [30]. Within the field of diagnostic development, microfluidic-based devices offer several key advantages, including precise fluid control, minimal reagent use, and analysis of extremely low analyte concentrations [31]. In addition, microfluidic devices can be self-contained and easily deployed as portable devices. These characteristics make microfluidics ideal for use in point-of-care (POC) platforms, where results are obtained relatively quickly during patient assessment.

In recent years, microfluidic devices have been constructed in attempts to improve the diagnosis of numerous infectious diseases; this includes scientists that have worked to develop microfluidic-based platforms for the diagnosis of Lyme disease. Nayak et al. (2016) presented a POC device for the detection of multiple anti-*B. burgdorferi* antibodies, known as the mChip-Ld (Figure 1a) [32]. The detection zones within the mChip-Ld device consist of Lyme antigens adsorbed onto the channel surface between red LED emitters and photodiodes (Figure 1b). When a sample containing the appropriate antibodies is passed over the zone, antibodies bind to the antigens to indicate the presence of *B. burgdorferi*. To detect antibody-antigen binding, a secondary gold-labeled antibody is added, which binds to the previous antibody. When a silver enhancer solution is added, a gold-catalyzed reduction of silver leads to a buildup of silver particles. Since the silver absorbs the wavelength emitted by the red LED, a proportional decrease in light intensity is observed when Lyme antibodies are present. After testing eight potential antigens, and their combinations, the OspC-K antigen returned promising results for early Lyme diagnosis, with a sensitivity of 84% and a specificity of 92%.

In an effort to improve the mChip-Ld test, Arumugam et al. (2019) tested 12 *B. burgdorferi* proteins to assess their corresponding antibody levels in early and late Lyme disease [33]. Of the tested proteins, three were selected for inclusion in the improved assay: VlsE, PepVF, and OspC. The improved assay obtained sensitivities of 80% and 85% with specificity set at 95% when tested on two early-Lyme serum panels; the STT algorithm applied to the same panels returned sensitivities of 48.5% and 75%. The observed increase in test sensitivity for patients with early Lyme disease shows promise for the eventual commercialization of a microfluidic-based assay that exceeds the STT’s capabilities.

### 2.2. Field Effect Transistors (FETs)

Field Effect Transistors (FETs) are a class of transistor, aptly named for their ability to control current through alterations in an electric field. Charge carriers, flowing from source to drain, are accelerated or slowed through a channel depending on the voltage applied across an adjacent gate region. FETs, particularly those utilizing carbon allotropes, provide functional biosensing platforms for the development of new diagnostic techniques [34].

While the application of FETs for Lyme sensing is a relatively new approach, there have been efforts to develop a carbon-based FET for Lyme disease diagnosis. Lerner et al. (2013) reported the functionalization of single-walled carbon nanotubes (SWCNT) with monoclonal antibodies for *B. burgdorferi* flagellar protein as transistors for an FET-based biosensor [35,36]. SWCNT were modified with diazonium carboxylates, which provided nanotube-localized carboxylic functionalities. From there, 1-ethyl-3-(3-dimethylaminopropyl) carbodiimide (EDC) and N-hydroxysulfosuccinimide (sulfo-NHS) were used as activators for the cross-linked addition of antibodies through available primary amine groups. The addition of flagellar protein antigen led to increases in the on-state current, indicating that antigen-antibody interactions tended to increase carrier mobility through the FET. Using this device, the group was able to achieve a limit of detection (LOD) of 1 ng/mL with only minutes of measurement required.

The group expanded upon their previous work with Gao et al. (2020), which sought to utilize graphene-based FETs (GFETs) for the simultaneous detection of four different *B. burgdorferi* antigens (P66, FlaB, GroES, and GroEL) [37]. To achieve detection, graphene was functionalized with single-chain variable fragment (scFv) antibodies using 1-pyrenebutyric acid and N-hydroxysuccinimide (NHS). These scFv consist of only the variable regions of the antibody heavy and light chains, making them much smaller than a typical antibody and allowing antibody-antigen interactions to occur much closer to the transistor. The completed biosensor consisted of an array of 100 GFETs, with each quadrant of sensors functionalized with one of four different scFv antibodies (Figure 2). By combining multiple biomarkers for multiplexed detection and using reduced antibodies, the device reached LODs between 2–500 pg/mL for each of the four biomarkers.

### 2.3. Lateral Flow Assays (LFAs)

Lateral Flow Assays (LFAs) are likely the most well-recognized type of biosensor in the world, owing mainly to home pregnancy tests—LFAs that have been on the market since the mid-1970′s. LFAs function via the passage of a liquid sample along various modified parts of a paper-based substrate. Typically, antigens bind to both free-floating label-conjugated antibodies and anchored antibodies further down the substrate—this indicates a positive antigen result. LFAs are inexpensive to manufacture, have long shelf-lives, and do not require refrigeration; these characteristics make them ideal for biosensor platforms, especially in developing areas [38].

While the current review highlights several biosensors with the potential to develop into commercial products, the only commercial Lyme biosensor to date is an LFA. The Sofia^®^ 2 Lyme fluorescent immunoassay (FIA), produced by Quidel^®^, detects both IgM and IgG antibodies to *B. burgdorferi* [39]. To operate the FIA, a 30 µL sample of blood serum/plasma is deposited into the device sample well and left to incubate. Once the sample has completed its migration across the test strip, the device analyzes the fluorescence and returns a positive, negative, or invalid result for both IgM and IgG. The device provides results within 3–15 min, can be stored/operated at room temperature, and has a shelf life of two years. When tested against a pre-existing Lyme enzyme immunoassay (predicate assay), the Sofia Lyme FIA showed higher IgM specificity (89.5% for FIA, 86.5% for predicate), slightly lower IgG specificity (96.5% for FIA, 98.5% for predicate), slightly higher IgM sensitivity (64.2% for FIA, 58.9% for predicate), and higher IgG sensitivity (80.0% for FIA, 49.5% for predicate). It should be mentioned that these comparisons were performed internally by Quidel^®^ [39].

### 2.4. Vertical Flow Assays (VFAs)

Vertical Flow Assays (VFAs) allow for the rapid detection of biological molecules by passing fluid samples vertically down through a molecule-coated substrate. This substrate is typically composed of individual or stacked paper-based layers and operates similarly to an LFA. VFAs offer several significant advantages over LFAs as the vertical flow of fluid leads to shorter assay times, no flow time requirement, and better multiplexing capability [40].

The development of VFAs for the rapid diagnosis of Lyme disease has been explored in recent years. Joung et al. (2019) presented a VFA for the multiplexed detection of three Lyme antibodies that could be analyzed using a mobile phone (Figure 3) [41]. One layer in the bottom case of the device—the sensing membrane—was impregnated with capture antigens (OspC, BmpA, P41) in 13 discrete reaction spots. Secondary gold nanoparticle-conjugated antibodies were deposited into a conjugation layer in the secondary top case. To run the device, a serum sample was injected into the primary top case (no conjugation layer) and washed through the device to the sensing membrane. After the sample antibodies had sufficient time to bind to the sensing membrane, the primary top case was swapped out for the secondary, and buffer was used to wash the gold-antibodies onto the sensing membrane. The color of the gold-antibodies could be seen with the naked eye, and a simple mobile phone camera was sufficient for intensity measurements. Overall, the device required only 20 min to produce results and obtained LODs of 209.6 ng/mL, 162.2 ng/mL, and 1.05 µg/mL for anti-OspC, anti-BmpA, and anti-P41, respectively.

More recently, Joung et al. (2020) improved upon their previous design by incorporating four additional antigens (DbpB, Crasp1, P35, Erpd/Arp37) and a synthetic peptide (Mod-C6) into the sensing membrane [42]. In addition, they used deep learning to train a neural network to determine an optimal diagnostic algorithm. Using the optimized algorithm on serum samples from patients with early-stage Lyme disease, the device achieved a sensitivity of 90.5% and specificity of 87.0%—this was improved to 85.7% and 96.3%, respectively, with threshold tuning and standardization. The results of this device show a significant improvement over current STT and MTT testing for early-stage Lyme and demonstrate the potential for an eventual rapid POC VFA test.

### 2.5. Surface Plasmon Resonance (SPR)

Surface Plasmon Resonance (SPR) spectroscopy is an analytical technique that uses reflected light to quantify the adsorption/binding of molecules onto a metal surface. As molecules accumulate on the metal surface, the angle of light required to induce electron resonance changes, allowing binding events to be easily quantified. Biosensors that employ SPR-related techniques are typically very sensitive, specific, fast, and represent some of the most advanced optical label-free biosensors available [43].

The use of SPR-based biosensing for Lyme disease was initially explored by Nagel et al. (2008), who developed a SPR sensor for *B. garinii* antibodies in human serum [44]. Carboxymethylated chips were functionalized with either OspC or VlsE antigens and/or their respective peptides (pepC10 and C6) using EDC and NHS. After sample addition and subsequent binding, the device achieved a sensitivity of 92% and specificity of 82% for the OspC/pepC10 pair and a sensitivity of 81% and specificity of 86% for the VlsE/C6 pair. The total analysis time for this SPR method was only a few minutes and required only 3 µL of patient serum, making it extremely fast and cost-effective.

### 2.6. Biochips

Biochips were born from the integration of microchips with biosensing technology and represented a significant advance in multiplexed sensing. Since each chip is composed of an array of individual biosensors, biochips are capable of measuring numerous analytes simultaneously in a high-throughput and parallel manner [45].

While biochips are a relatively new area of research, there have been some efforts to develop a functional biochip for Lyme disease. Huang et al. (2017) reported a biochip for the detection of *B. burgdorferi*, *B. garinii*, and *B. afzelii* that measured antibody responses to six *Borrelia*-specific antigens [46]. Of these six antigens, three were general to *Borrelia* species (flagellin, OspC, VlsE), and three were unique to each of the three species (VlsE IR6 peptides); this allowed for both confirmation of Lyme disease and differentiation between species. Antigens were immobilized onto gold slides in discrete reactive sites using succinimidyl undecanoate as a cross-linking agent. After exposure to potential anti-Lyme antibodies in serum, secondary fluorophore-conjugated antibodies were added to react with, and thus indicate the presence of, Lyme antibodies. These fluorescent antibodies were quantified using a fluorescent scanner, and a statistical threshold was used to determine Lyme status. Using this biochip, the authors obtained LODs of 0.39 µg/mL (anti-VlsE) and 0.78 µg/mL (anti-OspC and anti-flagellin).

More recently, Chou et al. (2020) presented a microfluidic protein microarray biochip (Figure 4) for the detection of antibodies against 16 different *B. burgdorferi* antigens [47]. Similar to the previous biochip, *Borrelia* antigens were immobilized onto a microchip—this time using a robotic microarrayer. After the antigen-coated microchip was exposed to serum samples, secondary fluorophore-labeled antibodies were used to indicate the presence of reactive anti-*Borrelia* antibodies. In addition, SPR was used to enhance the fluorophore signal using a method referred to as grating-coupled fluorescence plasmonics (GC-FP). Combinations of three, four, and five antigens were tested for their ability to determine Lyme status, where at least two positive antibody responses indicated *B. burgdorferi* presence. Using this threshold, the biochip reached peak sensitivities of 90% with 100% specificity, which outperformed the STT algorithm when applied to the same samples.

## 3. Summary and Future Outlooks

The development of biosensors for Lyme disease diagnosis is an area of emerging research and is set to see numerous advances over the next few years. While efforts to develop a Lyme biosensor date back to 2008 with the initial SPR-based biosensor [44], the last few years have seen the most intense progress in the field; this claim is supported by the observation that over half of the research articles reviewed here were published during or after 2019. This focus on improved Lyme diagnostics is fueled by both the increasing prevalence of the disease and the rapidly developing field of biosensor technology. Through this research, scientists aim to harness chemical and biochemical techniques to develop cheap, rapid, and effective tests for Lyme that can help stem the spread of the disease.

### 3.1. Achievements & Challenges

While the STT serology approach remains the most used method to diagnose Lyme disease, this standard dates back to 1994 [48] and has seen minimal improvement since then. A certain proverb may suggest that this methodology is fine as is—so long as it is not broken; however, while the test provides satisfactory results for those with disseminated Lyme, it struggles to diagnose those with early Lyme. The reported 46.3% sensitivity is considerably low, with a potentially large number of false-negative results within the first month of infection [15]. Likewise, the MTT approach demonstrates only slightly improved sensitivity, at approximately 50%, for patients with early Lyme disease [18]. Some of the biosensors presented here (Table 1) report sensitivities for early Lyme that exceed the STT/MTT approaches and have the potential to develop into a much-needed alternative [32,33,42]. However, some biosensors also demonstrate reduced specificity in comparison to the two-tiered approaches. Given the ambiguity of many of the Lyme disease symptoms, it is essential that any new sensor has high specificity to reduce false positives for other diseases.

In addition to their initial low sensitivity, the STT and MTT approaches are also quite expensive, both in analysis time (>24 h) and cost (>$400 per test) [42]. A rapid, inexpensive, and portable biosensor would prove to be an extremely beneficial tool, especially for rural areas where access to healthcare is limited. Several of the biosensors reviewed here have the potential to develop [37,47], or have already developed [39], into commercial Lyme sensors that approach or exceed the sensitivity and specificity of the STT and MTT methods. As well, some of the biosensors presented here utilize methods that are relatively inexpensive [39,41,42] in comparison to the STT/MTT approaches. These biosensors represent a substantial advance in the improvement of Lyme diagnostics and open the door to new inventions and innovations for both the diagnosis of Lyme disease and other infectious agents.

Lyme biosensors have the potential to lead to improved diagnostics; however, there are still challenges that must be overcome before they can provide reliable alternatives to STT/MTT testing. Deficient analyte concentration is one issue that often plagues the biosensor development process, especially those that target bacterial/viral antigens. While sensors may provide extremely accurate results for spiked samples, unless that success can be transitioned into real-world samples, the biosensor has limited applicability. Ideally, selected biomarkers should have high concentrations in the sample type (e.g., plasma), which can be approximated by bacterial count. *B. burgdorferi* cell estimates in various fluids/tissues span a wide range (Table 2); this variation can lead to difficulties in diagnosing certain individuals and thus, lower sensitivity. Besides appropriate sample selection, biomarkers should also be abundant (many per cell), highly specific (unique to the bacteria), and easily accessed (secreted/surface protein)—especially if the samples are not treated (e.g., lysed) prior to analysis.

Sensors that target antibodies, rather than bacterial antigens, risk experiencing similar issues to STT/MTT testing, with low sensitivity to early Lyme infection. Host-developed antibodies are excellent biomarkers as they are typically abundant, highly specific, and very stable. However, the time required for these markers to reach adequate detection levels may be invaluable to the patient’s health. The selection of alternative antibody targets and improvement of antibody visualization are just some of the potential steps that can be taken to counteract this delay.

While biosensors and other POC devices possess many advantages, including portability, inexpensiveness, and the decentralization of testing centers, there exist several issues that must be accounted for prior to wide-spread biosensor implementation. Decentralization of testing is great for expediting sample analysis time; however, it also leads to an increased risk of improper sample handling and analysis, including inadequate device calibration and insufficient quality control [53,54]. To address this, healthcare workers would need to undergo mandatory training or devices that would need to be simple enough, with detailed documentation, that they could be used by the consumer. While biosensors are less expensive in theory, some methods of biosensing require expensive analytic devices that would be impractical in an overly decentralized setting without a miniaturized version—which may be expensive to develop and produce. Biosensors may also utilize biological components that are highly susceptible to changes in temperature, pH, or other factors; this would make POC implementation difficult as results may vary depending on the environment where the analysis is taking place [55].

The development of new biosensors, both for Lyme disease and other infectious diseases, must include answers to the previously mentioned issues to have any hope of eventual commercialization. Since the ultimate goal of biosensors is typically deployment at the POC—either by health workers in a clinical setting or by oneself at home—the device must be thoroughly optimized, tested, and stable. This requires a thorough examination of potential factors that may influence the device’s outcomes, testing of the device with well-pedigreed, high-quality samples, and testing to ensure that the device is stable over a potentially long shelf-life. Scientists looking for examples of biosensor implementation need look no further than common home pregnancy tests and glucose sensing devices [56]. These biosensors began as in-lab assays and developed, through thorough testing and optimization, into wide-spread, inexpensive devices that are used by millions of people annually.

### 3.2. Looking Ahead

The current review represents the first consolidation of Lyme biosensor research to date, and functions to both summarize the field thus far and provide direction for future research. While several biosensing techniques have been explored so far, there exist many additional combinations of biorecognition elements and transducers that have shown high potential with the diagnosis of other infectious diseases. Optical methods (e.g., fluorescence, SPR) are the most common approach and are particularly good at high-throughput analysis [57]. Electrochemical methods (e.g., impedimetric, potentiometric) have shown great promise in developing sensors that are portable and easy-to-use—for example, glucose sensors for diabetes [58]. Piezoelectric methods (e.g., quartz crystal microbalance, acoustic wave), while not heavily commercialized as of yet, are excellent at effectively monitoring affinity interactions [59]. In addition to the transducing method, there also exists a wide array of potential recognition elements, including enzymes, aptamers, whole cells, and antibodies. Biosensors represent a subfield of chemistry that is both practical and continually evolving with the potential to solve the Lyme disease diagnostic dilemma.

## Figures and Tables

**Figure 1 biosensors-10-00137-f001:**
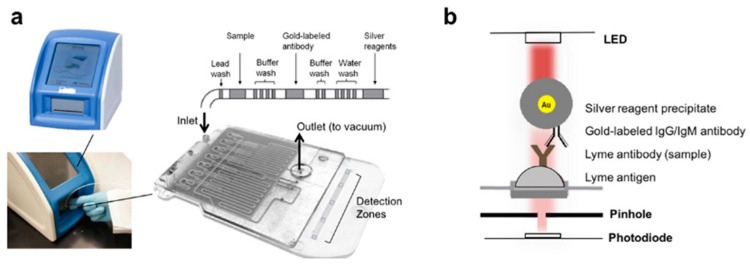
The mChip-Ld system showing (**a**) the microfluidic device and the order of buffers which are added to it, as well as the benchtop analyzer it is inserted into (**b**) the ordered binding of reagents and molecules within the detection zone. Reprinted from ref [32], which is open access under the Creative Commons Attribution 4.0 International license.

**Figure 2 biosensors-10-00137-f002:**
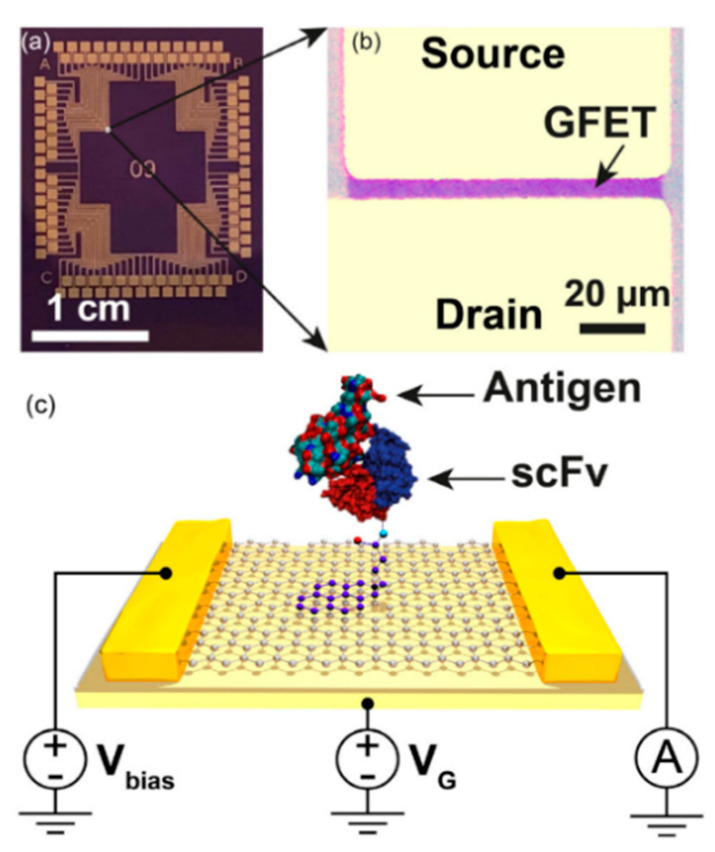
Graphene-based Field Effect Transistor (GFET) sensor showing (**a**) sensor array containing 100 GFET units, (**b**) optical micrograph of a single unit (**c**) visualization of the GFET with attached scFv binding to antigen. Reprinted from ref [37], with permission from IOP Publishing.

**Figure 3 biosensors-10-00137-f003:**
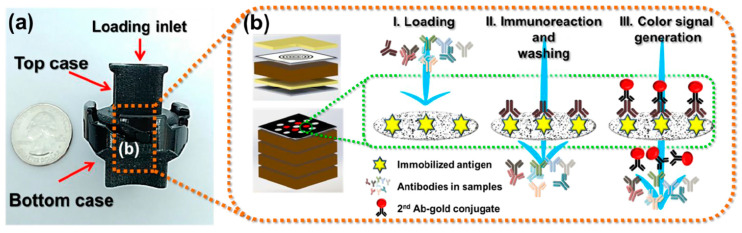
Vertical Flow Assay (VFA) device showing (**a**) assembled device relative to an American quarter (**b**) the general assay procedure, from sample addition to eventual sample color generation. Reprinted from ref [41], with permission from the Royal Society of Chemistry.

**Figure 4 biosensors-10-00137-f004:**
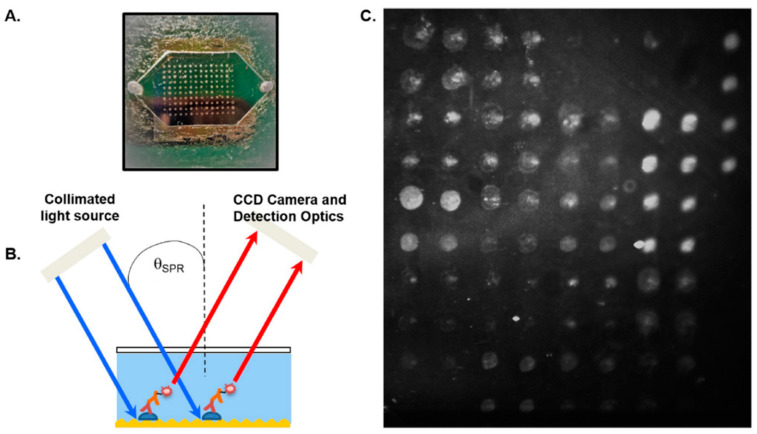
Microfluidic protein biochip showing (**A**) biochip device with overlying microfluidic window (**B**) illustration of antigen-antibody binding process with Surface Plasmon Resonance (SPR)-enhanced fluorescence (**C**) fluorescent image of an antibody-bound biochip. Reprinted from ref [47], which is open access under the Creative Commons Attribution 4.0 International license.

**Table 1 biosensors-10-00137-t001:** Comparison of reported Lyme disease biosensors. Bold font represents biosensors tested specifically against early Lyme disease samples.

Method	Sample	Target	Sensitivity or LOD	Specificity	Time	Ref.
Microfluidics	**Serum**	**Antibodies**	**84%**	**92%**	**15 min**	[32]
	**Serum**	**Antibodies**	**80%/85%**	**95%**	-	[33]
FET	Spiked Buffer	Antigen	1 ng/mL	High	Minutes	[35]
	Spiked Buffer	Antigen	2–500 pg/mL	High	-	[37]
LFA	Serum	Antibodies	64.2%/80%	89.5%/96.5%	3–15 min	[39]
VFA	Serum	Antibodies	162.2–1046 ng/mL	-	20 min	[41]
	**Serum**	**Antibodies**	**85.7%**	**96.3%**	**15 min**	[42]
SPR	Serum	Antibodies	81%/92%	86%/82%	A few min	[44]
Biochip	Serum	Antibodies	0.39–0.78 µg/mL	-	-	[46]
	Serum	Antibodies	90%	100%	-	[47]

LOD = Limit of Detection, FET = Field Effect Transistor, LFA = Lateral Flow Assay, VFA = Vertical Flow Assay, SPR = Surface Plasmon Resonance.

**Table 2 biosensors-10-00137-t002:** Estimates of *B. burgdorferi* cell count in various clinical sample types.

Sample Type	Range	Reference
Plasma	<20 to >4000 cells/mL	[49]
Urine	<50 to >5000 cells/mL	[50]
Synovial Fluid	20 to 41,000 cells/mL	[51]
Skin Biopsy	10 to 11,000 cells per 2 mm	[52]
Cerebral Spinal Fluid	63 cells/mL	[51]

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
