# Peer review of "Lyme Disease Biosensors: A Potential Solution to a Diagnostic Dilemma"

_biosensors, 2020, doi:10.3390/bios10100137_

Round 1
Reviewer 1 Report
in attachment

Reviewer 2 Report
I consider the document to be well written, with no English mistakes and pleasant writing for the reader, besides high scientific rigor. This review stands out from the literature due to its comprehensiveness in the biosensors theme, being explored new biosensing techniques such as microfluidics, field-effect transistors, lateral flow assays, vertical flow assays, surface plasmon resonance, and biochips. The authors also gave their opinion about the current and future panorama in the diagnostic methods of this disease, highlighting the main achievements & challenges. As for the bibliography used, this was adequate and current.
However, the manuscript should be improved and minor revisions should be addressed before accepting this manuscript in Biosensors. Some detailed comments are list as follows.
- Line 45: The reference that supports the percentage of cases that present rash in Lyme diseases is not the most appropriate. The authors should replace it. (suggestion: Skar GL, Simonsen KA. Lyme Disease. [Updated 2020 Jul 10]. In: StatPearls [Internet]. Treasure Island (FL): StatPearls Publishing; 2020 Jan-.)
- Line 57: Please include the following reference: Mead P, Petersen J, Hinckley A. Updated CDC Recommendation for Serologic Diagnosis of Lyme Disease. MMWR Morb Mortal Wkly Rep 2019;68:703. DOI: http://dx.doi.org/10.15585/mmwr.mm6832a4.
- In section 2. The authors highlight the advantages of biosensors, however, the major disadvantages of the use of these diagnostic methods should be presented.
- In section 2.1. the authors should explore the use of portable microfluidic devices for Lyme disease biosensing.
- A table should be included in order to compare the different types of biosensors used regarding their limit of detection (LOD), sample type used, detection time, major advantages/ disadvantages.
- In table 1: Please confirm the range for urine and blood samples.
- There is no patent cited in this review, I suggest that authors look for patents that have been developed for Lyme disease biosensing.
- An economic analysis of the development cost of the different types of biosensors should be included in the review.
Round 2
Reviewer 1 Report
I am very impressed with the revision and the authors have addressed each of my comments in a thoughtful and complete manner. As a result, I have no issues with the current state of the manuscript and it is ready for publication from my perspective.